**Data Availability Statement:** All relevant data are within the manuscript and its Supporting Information files.

**Funding:** JLC received funding from Agencia Nacional de Promoción Científica y Tecnológica

# A graph theory approach to analyze birth defect associations

**Dario Elias**[1,2], **Hebe Campaña**[1,2,3], **Fernando Poletta**[1,2,4], **Silvina Heisecke**[1], **Juan Gili**[1,2,5], **Julia Ratowiecki**[1,2], **Lucas Gimenez**[1,2,4], **Mariela Pawluk**[1,2], **Maria Rita Santos**[1,2,3,6], **Viviana Cosentino**[1,2], **Rocio Uranga**[1,2,7], **Monica Rittler**[1,2,8], **Jorge Lopez Camelo**[1,2,4]*

1 Laboratorio de Epidemiología Genética, Centro de Educación Médica e Investigaciones Clínicas-Consejo Nacional de Investigaciones Científicas y Técnicas (CEMIC-CONICET), Ciudad Autónoma de Buenos Aires, Argentina, 2 Estudio Colaborativo Latino Americano de Malformaciones Congénitas, CEMIC-CONICET, Ciudad Autónoma de Buenos Aires, Argentina, 3 Comisión de Investigaciones Científicas, Buenos Aires, Argentina, 4 Instituto Nacional de Genética Médica Populacional, CEMIC-CONICET, Ciudad Autónoma de Buenos Aires, Argentina, 5 Instituto Académico Pedagógico de Ciencias Humanas, Universidad Nacional de Villa María, Córdoba, Argentina, 6 Instituto Multidisciplinario de Biología Celular, Buenos Aires, Argentina, 7 Hospital San Juan de Dios, Buenos Aires, Argentina, 8 Hospital Materno Infantil Ramón Sarda, Buenos Aires, Argentina

* jslc@eclamc.org

## Abstract

Birth defects are prenatal morphological or functional anomalies. Associations among them are studied to identify their etiopathogenesis. The graph theory methods allow analyzing relationships among a complete set of anomalies. A graph consists of nodes which represent the entities (birth defects in the present work), and edges that join nodes indicating the relationships among them. The aim of the present study was to validate the graph theory methods to study birth defect associations. All birth defects monitoring records from the Estudio Colaborativo Latino Americano de Malformaciones Congénitas gathered between 1967 and 2017 were used. From around 5 million live and stillborn infants, 170,430 had one or more birth defects. Volume-adjusted Chi-Square was used to determine the association strength between two birth defects and to weight the graph edges. The complete birth defect graph showed a Log-Normal degree distribution and its characteristics differed from random, scale-free and small-world graphs. The graph comprised 118 nodes and 550 edges. Birth defects with the highest centrality values were nonspecific codes such as *Other upper limb anomalies*. After partition, the graph yielded 12 groups; most of them were recognizable and included conditions such as VATER and OEIS associations, and Patau syndrome. Our findings validate the graph theory methods to study birth defect associations. This method may contribute to identify underlying etiopathogeneses as well as to improve coding systems.

## Introduction

Birth defects (BD) are prenatal morphological or functional anomalies, classified as major or minor according to their clinical or biological significance. BD often exert a significant effect

(ANPCyT) PICT 2016-0952. The funders had no role in study design, data collection and analysis, decision to publish, or preparation of the manuscript.

**Competing interests:** The authors have declared that no competing interests exist.

on the newborn's health as well as burden on treatment resources. The estimated BD prevalence, which has not substantially changed over time, is around 3% worldwide [1,2].

Since the thalidomide episode, BD surveillance has become a public health concern. BD registries have been created to identify new teratogens and studies, focusing on BD associations, have been carried out [3–7]. BD associations are defined as the unknown etiopathogenesis coexistence of two or more unrelated anomalies.

Mainly two approaches have been used to analyze associations between unrelated BD. One of them focuses on a specific defect and determines its association degree to other anomalies in combinations of two, three, or four. For this approach, observed versus expected ratios, and multivariate methods such as log-linear models have been used [8–10]. The second approach is based on clustering methods. This method codes each newborn BD using a binary rating system, and then newborns are clustered into groups. Discriminant analyses are performed to identify the anomalies significantly associated to each group [11,12].

Graph theory may be considered as another approach to analyze BD associations. A graph consists of nodes which represent the entities (BD in the present work), and edges that join nodes indicating the relationships among them [13]. For this approach, a vast amount of clustering algorithms have been designed [14], such as Infomap, which is based on the graph information flow [15]. Centrality measures have been designed to characterize each node according to its associations and those of its neighbors. For example, *Degree* represents the number of edges leading to a node, *Betweenness* evaluates the number of shortest paths that pass through a node, and *Eigenvector* considers the number of associations of a node, as well as those of the nodes it is connected to [13]. Thereby, the graph theory methods allow to focalize the whole set of BD, to characterize each BD according to its centrality value, and to identify groups.

The graph theory approach has been applied to social networks allowing to identifying influencers [16], and the propagation and impact of fake news [17]. Further networks have focused on protein interactions leading to identify the human interactome [18], genetic interactions defining the cell map of yeasts [19], and association between diseases with common genetic mutations [20], or common symptoms [21]. However, to our knowledge, the graph theory approach has not yet been applied to analyze BD associations. Although experimental as well as epidemiologic studies have been and are still being carried out to unravel pathogenic paths underlying BD associations, most of them have been unsuccessful [22].

The aim of the present study was to verify the ability of the graph theory methods to identify already known BD associations, and thereby to consider its inclusion as a further tool used in BD surveillance.

## Material and methods

### Ethical aspects

The study protocol was approved by the Ethics Committee "Centro de Educación Médica e Investigaciones Clínicas (CEMIC)" (DHHS-IRB #1745, IORG #1315). Written and signed informed consents are obtained from all subjects participating in the Estudio Colaborativo Latinoamericano de Malformaciones Congénitas (ECLAMC) program before data collection. Furthermore, ECLAMC pediatricians adequately explain the written informed consent content to the mother or legal guardian of the newborn. All data were fully anonymized prior to their utilization. All written consents are available in the ECLAMC coordination headquarters.

### Data collection

For the present work, ECLAMC BD database was used. ECLAMC is a hospital-based BD monitoring system that has been operating in twelve Latin American countries since 1967 [23]. It

records all major and minor anomalies diagnosed at birth or before the infant's hospital discharge. Between 1967 and 2017, around 5 million live and stillborn infants have been examined; 170,430 cases (around 3%) had one or more BD.

Written and signed informed consents are obtained from all subjects participating in the ECLAMC program before data collection. Furthermore, ECLAMC pediatricians adequately explain the written informed consent content to the mother or legal guardian of the newborn.

The ECLAMC BD classification system combines ICD8 and specific ECLAMC codes for major and minor anomalies, as well as for some chromosome anomaly syndromes.

For the present work, codes recorded in the whole ECLAMC BD set of live and stillborn infants were used.

## Representation of BD associations as a graph

To study BD associations using the graph theory, anomalies were represented as nodes and their associations as undirected weighted edges. Edges were weighted using the association strength volume-adjusted Chi-Square (VA-Chi2) (S1 Appendix). VA-Chi2 can be interpreted as a distance from independence; a value close to zero indicates that events are independent [24].

The number of edges in the BD graph (BDG) was defined based on a minimum number of cases with two defects (I parameter = 18), and considering the edges with the strongest association (VA-Chi2) (A parameter = 550). Parameters I and A were selected based on the average codeword length described in S1 and S2 Figs.

## Graph partition

Graph partition was performed with the Infomap method (version 0.19.21) [15]. Nodes could only belong to one group of the obtained in the partition (Infomap default setting). The Infomap algorithm is based on the information flow tendency within well connected groups. Groups composing a network are identified by the definition of an optimally compressed description of the way information flows in the network. The Huffman code [25] is used to describe the information flow through an infinite random walk, considering it as a proxy of the flow in the network.

Two measures were used to evaluate the graph partition quality:

**Modularity.**   It reflects the concentration of edges within groups compared to a random distribution of edges; its values range between -1 and 1. The closer to 1, the higher is the partition quality [26].

**Average codeword length.**   It is an information-theoretical approach that uses a measure based on the Huffman code length of a random walker whose values are greater than zero. The lower the value, the higher is the partition quality [15].

**Network characteristics.**   The following metrics were analyzed [27]:

**Density (D).**   Density of a graph indicates the number of associations between nodes. It is calculated as the number of observed associations over the number of all possible associations:

D = edge number x 2/nodes x (nodes-1)

D values range between 0 and 1.

**Degree assortativity (DA).**   It represents the Pearson correlation coefficient of degree between pairs of linked nodes. Its values range between -1 and 1 [28]. Nodes with values closer to 1 tend to associate to nodes with similar degree values; those with low values tend to associate to nodes with different numbers of associations.

**Clustering coefficient (CC).**   It measures the connection degree among neighbors of a node. In a graph, it is calculated as the number of triplets of fully connected nodes over all possible triplets. It ranges between 0 and 1, where 1 indicates a fully connected graph.

**Average short path length (ASPL).** It indicates the average of the minimum number of steps (edges) among all node pairs.

**Small-World index.** Small-World is a kind of graph. Telesford et al. (2011) designed an index to measure the belonging of a graph to the Small-Word kind [29]. The index compares network clustering to an equivalent lattice network, and path length to a random network. Its values range from -1 to 1, where -1 indicates that the graph has features of a lattice graph, zero means that it has characteristics of a Small-World graph, and 1 that it has appearances of a random graph.

To characterize BD, based on their position in the graph, three centrality indexes were used:

**Degree.** Number of edges of a node.

**Strength.** Sum of edge weights of a node.

**Eigenvector (EV).** Its values arise from a reciprocal process where the centrality of each node is proportional to the sum of centralities of the nodes it is connected to. Algebraically, the EV centrality refers to the values of the first EV of the graph weighted with the adjacency matrix. A normalized EV value between 0 and 1 was used. Nodes with values close to 1 indicate a high number of associated nodes as well as its connection to nodes with a high number of associations.

Validation of the results was based on clinical evaluation and interpretation of BD groups obtained in the partitioned graph.

## Results

The complete BDG comprised 118 nodes and 550 edges (S1 and S2 Tables). Its edge density was 0.08, and its degree and EV centralities were 0.28 and 0.78, respectively. This BDG differed from the three models taken as reference (Table 1). The median BDG values (CC, DA, and ASPL) were not within the 95% confidence interval of the random graphs generated with Erdos & Renyi (ER) [30], and Barabási & Albert (BA) [31] models, nor within random graphs generated with the same degree and weight distributions (SDD); meanwhile, the Telesford et al. (2011) small-world index for the BDG was -0.30. BDG degree distribution was closer to a Log-Normal distribution (Kolmogorov-Smirnov distance: 0.05, P-Value: 0.23) than to other distributions such as Power and Poisson (S4 Table).

BDG partition yielded 12 groups (Fig 1, S3 Table). Ten groups represented known BD associations, syndromes, or clinically consistent complexes, while two of them (groups 9 and 10) did not.

Nodes that showed the highest degree, EV and strength values were *Other upper limb anomalies* and *Microretrognathia* (S5 Table). The three strongest associations were *Localized edema* with *Turner syndrome*, *Proboscis* with *Cyclopia*, and *Liver and bile duct defects* with *Other spleen defects* (S1 Table).

**Table 1. Comparison between the birth defects graph and three graph models.** Erdos & Renyi, Barabási & Albert, and Same Degree Distribution data correspond to the median of 1000 graphs. The models were created with the same number of nodes, edges, and weight distribution as the birth defects graph.

| Features | Birth Defect Graph | Erdos & Renyi | Barabási & Albert | Same Degree Distribution |
|---|---|---|---|---|
| Clustering Coefficient | 0.41 | 0.08 | 0.21 | 0.30 |
| Degree Assortativity | -0.05 | -0.02 | -0.29 | -0.12 |
| Average shortest path length | 2.72 | 2.36 | 2.31 | 2.50 |
| Modularity | 0.32 | 0.19 | 0.06 | 0.07 |
| Average codeword length | 5.28 | 6.31 | 6.04 | 6.03 |

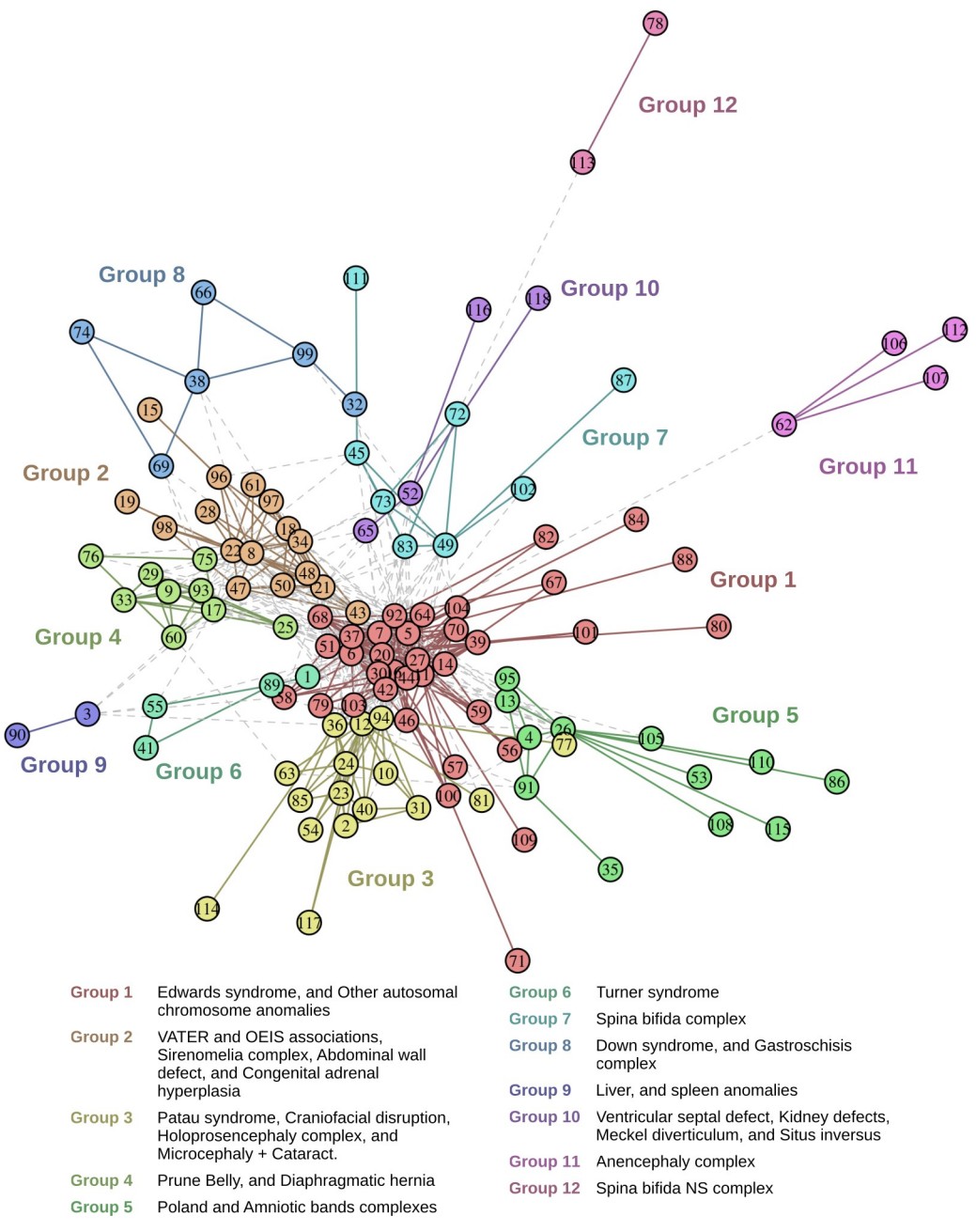

**Fig 1. Birth defects graph.** Each node represents a birth defect code. S2 Table depicts code names. Color of nodes and edges indicates the partition group to which they belong. Edges between groups are gray and dotted.

Minor BD presented a median EV centrality of 0.10, greater than the observed for major BD (0.03) (Wilcoxon p-value 0.0004). Minor BD also presented greater degree (7.50 vs. 5.00) and strength (0.60 vs. 0.44) values but their significance was lower than major BD values (Wilcoxon p-value 0.0559 and 0.0396, respectively).

Group 1 showed the highest number of nodes (35), 62% were minor BD; it included syndromes such as Edwards and *Other autosomal chromosome anomalies* (Fig 1).

Following, Groups 2, 3, and 5 are described as examples.

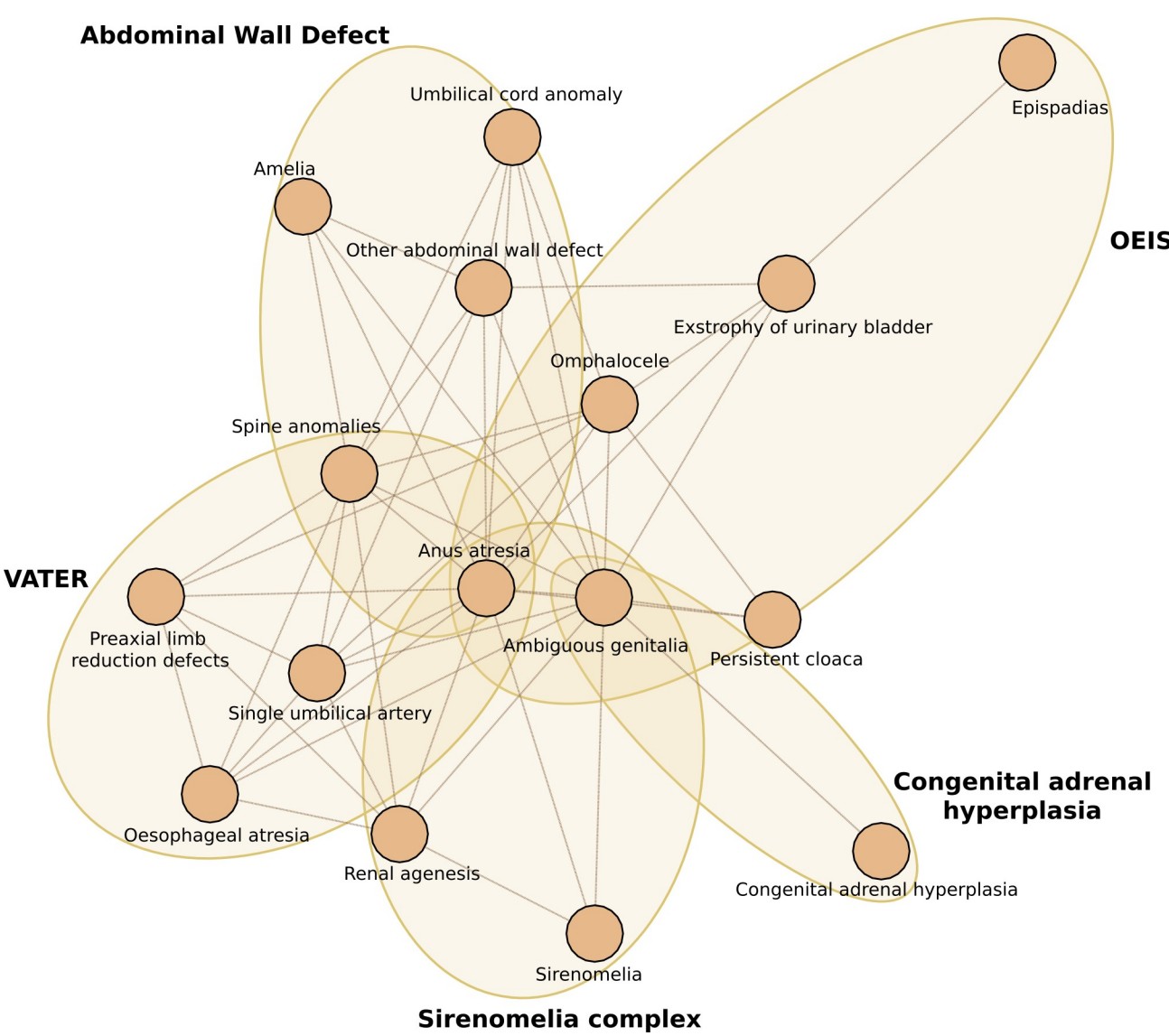

**Fig 2. Subgroups and associated anomalies identified in Group 2.**

Group 2, shown in Fig 2, comprised 16 nodes and 50 edges; its density, degree, EV, and CC centrality values were 0.41, 0.45, 0.50, and 0.59, respectively. Nodes with the highest degree and EV values were *Ambiguous genitalia*, *Anus atresia*, and *Spinal anomalies* (S3 Table). The three strongest associations were *Ambiguous genitalia* with *Anus atresia*, *Epispadias* with *Exstrophy of urinary bladder*, and *Other abdominal wall defect* with *Umbilical cord anomaly* (S1 Table). The following subgroups could be identified: VATER and OEIS associations, Sirenomelia complex, *Abdominal wall defect* (which included the Limb-body wall-complex and Short cord), and *Congenital adrenal hyperplasia*.

Group 3, shown in Fig 3A, comprised 16 nodes and 40 edges; its density, degree, EV and CC centrality values were 0.33, 0.53, 0.57, and 0.51, respectively. Nodes with the highest degree values were *An/Microphthalmia* and *Holoprosencephaly*; those with the highest EV values were Patau Syndrome and *An/Microphthalmia* (S3 Table). The three strongest associations were *Proboscis* with *Cyclopia*, *Aplasia cutis vertex* with Patau syndrome, and *An/Microphthalmia*

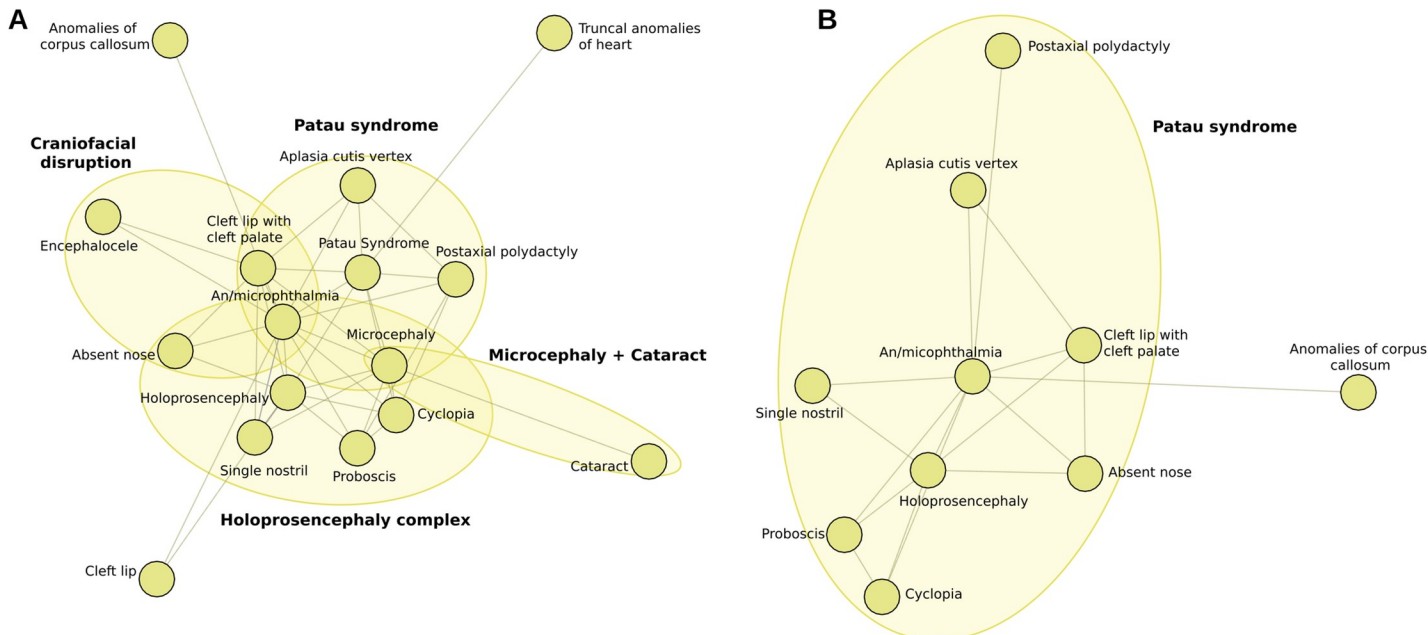

**Fig 3. Subgroups and associated anomalies identified in Group 3.** (A) With Patau syndrome code. (B) Without Patau syndrome code.

with Patau syndrome (S1 Table). The following subgroups could be recognized in Group 3: Patau syndrome, *Craniofacial disruption*, Holoprosencephaly complex, and *Microcephaly +Cataract*. When a graph was created excluding the Patau syndrome code (with parameters I = 18 and A = 350), most of the previously found BD still appeared while others, such as *Truncal heart anomalies*, *Encephalocele*, *Microcephaly*, *Cataract*, and *Cleft lip without cleft palate*, did not (Fig 3B).

Group 5, shown in Fig 4, comprised 12 nodes and 15 edges; its density, degree, EV and CC centrality were 0.23, 0.68, 0.70, and 0.20, respectively. The node with the highest degree value was *Syndactyly*, while those with the highest EV values were *Constriction band scar*, and *Amputation* (S3 Table). The three strongest associations were *Constriction band scar* with *Amputation*, *Limb hypoplasia* with *Pectoralis muscle defect*, and *Limb hypoplasia* with *Syndactyly* (S1 Table). Two subgroups (Poland and Amniotic band complexes) could be identified. *Syndactyly* also independently associated to each one of a number of *Limb reduction defects* and *Polydactylies*.

## Discussion

The usual way to characterize a graph is by comparing it with other networks and models to identify specific features of the phenomenon under study [32].

In the present work, one of the compared features was degree distribution. The BDG showed a better adjustment to a Log-Normal distribution, which is in accordance with Broido & Clauset (2019) who observed this distribution in most empirical networks [33]. However, the BDG differed from ER graphs widely used as the backbone of null models [34], and whose degree distribution follows a Poisson law [30]. This difference with ER graphs suggests that the association probability between two BD is not constant. Furthermore, the BDG Log-Normal distribution also determined its difference from scale-free graphs whose distribution follows a Power law. Scale-free graphs are characterized by the presence of few nodes with a degree that greatly exceeds the average (i.e. World Wide Web network) [35]. Finally, small-world graphs

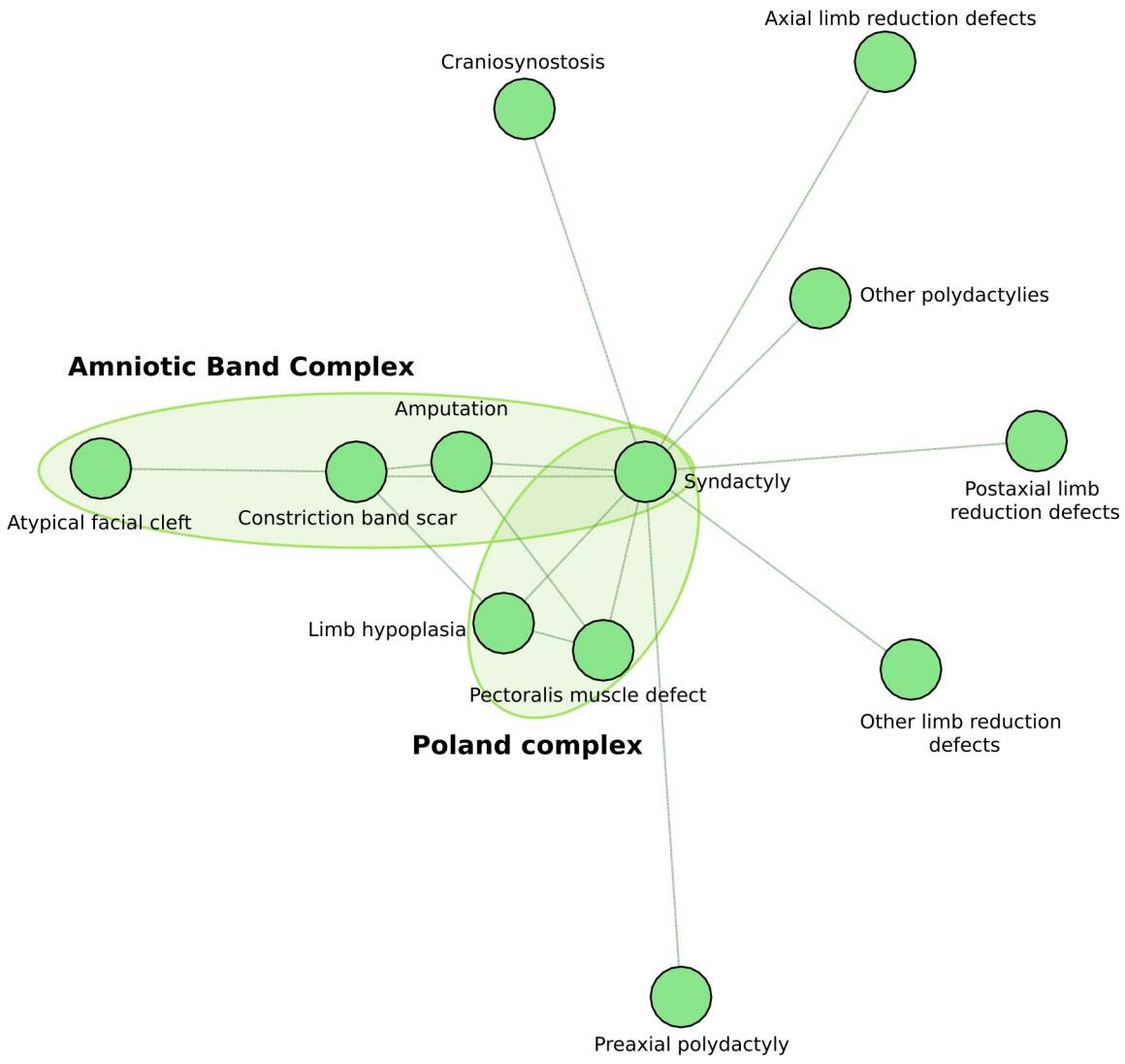

**Fig 4. Subgroups and associated anomalies identified in Group 5.**

present a high clustering coefficient and a small characteristic path length implying the presence of sub networks and a global reachability property [36]. Many real networks meet these characteristics [37–39], while the BDG has differed.

That the BDG did not adjust to any of these models is an indicator of its singularity which would require further studies of the BD registry characteristics.

## Graph partition

The graph partition showed a modularity greater than zero and greater than the observed in ER and SDD graphs, as well as a lower code length. These characteristics suggest a modular structure where each group (or module) could associate to different clinical conditions. That several related conditions were detected within each group is in accordance with this observation.

In group 2, at least five overlapping conditions could be recognized, with *Anus atresia* acting as the link among them. Although *Ambiguous genitalia* also showed high centrality values,

some of its associated BD, such as a *Persistent cloaca* or *Congenital adrenal hyperplasia*, are pathogenetically related and therefore redundant.

In Group 3, the highest centrality values for *An/Microphthalmia* point to this defect as the most common one to the observed overlapping conditions (meaning the strongest link among them), and probably to other outlying groups as well. It was followed by *Holoprosencephaly*, which appeared as the link between BD related to the Holoprosencephaly complex (*Single nostril*, *Absent nose*, *Proboscis*, and *Cyclopia*) and Patau syndrome.

In group 5, two overlapping conditions (Poland and Amniotic band complexes) could be recognized linked by *Syndactyly*. The observed association between *Syndactyly* and *Craniosynostosis* probably represents a group of syndromes known as *Acrocephalopolysyndactylies*.

The congruence between obtained BDG groups and clinical conditions demonstrates the ability of the graph theory approach to identify known associations. This ability is further highlighted by the fact that even after excluding the Patau syndrome code in Group 3, the obtained defects still suggested this diagnosis. The differences between groups with and without the Patau syndrome code may have a biological meaning; however, they could also be due to operational factors. For example, when justifying the diagnosis of a Patau syndrome, the operator should be very exhaustive and describe redundant BD such as *Microcephaly*, which otherwise, he/she would not. This, in turn, would lead to the appearance of defects these redundant anomalies are associated to, such as *Encephalocele* or *Cataracts*.

The application of the graph theory methods to so far undefined case registries with multiple BD could help establish new associations.

## Minor anomalies

While an association is defined as the coexistence of two or more mainly major independent BD (i.e. VATER, OEIS, etc.), a syndrome refers to a known or suspected condition, with or without major anomalies, whose recognition often mainly relies on minor BD (i.e. Down or Edwards syndromes).

The group with the highest number of nodes was the one including syndromes such as Edwards and other autosomal chromosome anomalies and, as expected, 62% of them were minor BD.

Minor anomalies often present alone as well as in association with a number of different conditions [40]. That in our sample minor anomalies showed the highest EV values could indicate a preferential reporting of such BD when they occur in association to others.

## Unspecified defects

Some ill-defined BD such as *Other upper limb anomalies* or *Neck anomalies*, which often associate to other conditions, were also among those with the highest EV values. That the EV values of some equally unspecific BD, such as *Liver and bile duct anomalies* or *Other adrenal defects* were low could be due to the low overall prevalence (or low detection rate) of such BD in recognized associations.

Therefore, indexes such as EV could be used to improve coding systems by detecting codes whose low specification level might interfere when interpreting the network.

## Limitations

The present results were mainly determined by the established significance thresholds (I and A parameters), which were selected after evaluating the graph partition quality, as well as by the recognition of known associations. The latter was prioritized, given the exploratory nature of

this work; therefore, lax thresholds were selected. However, even when using tighter thresholds, known clinical conditions could still be identified.

## Conclusions

The findings of this work suggest the graph theory as a new approach to study BD associations, as well as a tool to evaluate and improve coding systems. Its main advantage is the ability to analyze relationships among defect complexes and associations which may lead to the identification of common pathways and, eventually, to their etiopathogeneses. With this aim, a number of studies are being planned which could identify associations that have not yet been described, and may add information to those already known.

## Supporting information

**S1 Fig. Median average codeword length by minimum number of cases with both birth defects.** The median average codeword length corresponds to the partition of weighted graphs generated with the VA-Chi2 function and different threshold values: I) minimum number of cases with both defects between 10 and 30, with a step of 1; A) number of edges (with greater strength of association) included in the graph, between 50 and 800 with a step of 25. The blue dot was the threshold selected in this work.
(TIF)

**S2 Fig. Variation of average codeword length (ACL) by number of edges.** The average codeword length corresponds to the partition of weighted graphs generated with the VA-Chi2 function, a minimum number of cases with both defects of 18, and number of edges (A) (with greater strength of association) included in the graph, between 50 and 800 with a step of 25. The variation of ACL for each value of A was calculated with respect to the previous value of A (ordered from highest to lowest). A positive variation indicates a decrease in the ACL. The blue dot was the threshold selected in this work.
(TIF)

**S1 Table. Edges included in the birth defects graph.** Chi2: Chi-Square independence test. VA-Chi2: volume-adjusted Chi-Square independence test. Group: Partition group, edge between groups has empty value.
(XLSX)

**S2 Table. Description of birth defects codes.**
(XLSX)

**S3 Table. Nodes of each partition group and their centrality indices with respect to their group.**
(XLSX)

**S4 Table. Function adjustment to the degree distribution of the graph.** KS Dist: Kolmogorov-Smirnov distance. X min: Initial grade where adjustment begins.
(XLSX)

**S5 Table. Node centrality indices with respect to complete graph.**
(XLSX)

**S1 Appendix. Methodological details.**
(DOCX)

## Acknowledgments

The authors want to thank all physicians collaborating in the ECLAMC network, and Mariana Piola and Alejandra Mariona for technical assistance. Also L. A. Spinetta.

## Author Contributions

**Conceptualization:** Dario Elias, Fernando Poletta, Lucas Gimenez, Monica Rittler, Jorge Lopez Camelo.

**Data curation:** Dario Elias, Hebe Campaña.

**Formal analysis:** Dario Elias, Hebe Campaña, Monica Rittler, Jorge Lopez Camelo.

**Funding acquisition:** Jorge Lopez Camelo.

**Investigation:** Lucas Gimenez, Jorge Lopez Camelo.

**Methodology:** Dario Elias, Fernando Poletta, Juan Gili, Jorge Lopez Camelo.

**Project administration:** Jorge Lopez Camelo.

**Resources:** Jorge Lopez Camelo.

**Supervision:** Jorge Lopez Camelo.

**Validation:** Dario Elias, Fernando Poletta, Monica Rittler.

**Visualization:** Dario Elias, Fernando Poletta, Juan Gili, Lucas Gimenez, Mariela Pawluk, Viviana Cosentino, Monica Rittler, Jorge Lopez Camelo.

**Writing – original draft:** Dario Elias, Hebe Campaña, Juan Gili, Julia Ratowiecki, Lucas Gimenez, Mariela Pawluk, Maria Rita Santos, Viviana Cosentino, Rocio Uranga, Monica Rittler, Jorge Lopez Camelo.

**Writing – review & editing:** Silvina Heisecke.

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
