## [Decision Letter · Decision Letter 0]

25 Feb 2020

PONE-D-19-35846

The Graph Theory: A new approach to analyze birth defect associations

PLOS ONE

Dear Dr López-Camelo,

Thank you for submitting your manuscript to PLOS ONE. After careful consideration, we feel that it has merit but does not fully meet PLOS ONE’s publication criteria as it currently stands. Therefore, we invite you to submit a revised version of the manuscript that addresses the points raised during the review process.

We would appreciate receiving your revised manuscript by Apr 10 2020 11:59PM. To enhance the reproducibility of your results, we recommend that if applicable you deposit your laboratory protocols in protocols.io, where a protocol can be assigned its own identifier (DOI) such that it can be cited independently in the future. For instructions see: http://journals.plos.org/plosone/s/submission-guidelines#loc-laboratory-protocols

We look forward to receiving your revised manuscript.

Kind regards,

Diego Raphael Amancio

Academic Editor

PLOS ONE

Journal Requirements:

2. In ethics statement in the manuscript and in the online submission form, please provide additional information about the database used in your retrospective study. Specifically, please ensure that you have discussed whether all data were fully anonymized before you accessed them and/or whether the IRB or ethics committee waived the requirement for informed consent. If patients provided informed written consent to have their data used in research, please include this information.

Reviewers' comments:

Reviewer's Responses to Questions

**Comments to the Author**

1. Is the manuscript technically sound, and do the data support the conclusions?

Reviewer #1: Partly

2. Has the statistical analysis been performed appropriately and rigorously? 

Reviewer #1: Yes

3. Have the authors made all data underlying the findings in their manuscript fully available?

Reviewer #1: No

4. Is the manuscript presented in an intelligible fashion and written in standard English?

Reviewer #1: No

5. Review Comments to the Author

Reviewer #1: In this manuscript the authors apply graph theoretic methods to the analysis of patterns of birth defects diagnoses in a large collaborative birth defects surveillance database. The paper holds some interest, but should be framed in a broader context and substantially revised. Some specific comments appear below.

First, the title and throughout refers to 'The Graph Theory'. Actually what the authors do is to apply graph-theoretic methods to analysis of birth defects diagnoses. There are other approaches under exploration that take a similar approach, using machine learning methods. The authors should discuss their approach within this context, and describe how it differs and what advantages if any it offers. All of these approaches are agnostic in that they don't take into account things we might know about diagnoses, so the value-add from the approach outlined here should be interpreted in context as well.

Second, throughout the text there are poorly constructed sentences, mis-spelling, statements that are not as well referenced as they might be. Taking just the abstract and first page of the body of the manuscript as examples:

In the abstract, all acronyms should be spelled out, and the actual acronym used only if referenced more than once. Saying the findings 'seems to validate' the proposed method is a bit weak.

On p 3, line 54, one of these references is from 1983?

line 55 - what is meant by thalidomide and rubella 'episodes'? Thalidomide has been strongly linked with the development of birth defects surveillance, and there is a literature to support this (none cited), but this is less true for congenital exposure to Rubella, which came to attention decades before any substantial activity in birth defects surveillance ensued.

line 59, would read better as 'Most studies have used one of two approaches to analyze associations between unrelated birth defects.' Then, 'One approach focuses . . .'

lines 60-62, this sentence seems to over-generalize

line 62, you mean 'multivariable' rather than 'multivariate'

line 73, should use 'have' rather than 'has' to refer to 'vast amount'

The above comments refer to a single page in the manuscript. This reviewer could make similar recommendations throughout, and suggests that the entire manuscript be copyedited by a scientist familiar with birth defects surveillance who has English as a first language.

Additional comments:

p 4 line 91-94: it doesn't necessarily follow that, if the graph-theoretic approach works well to identify known associations it will therefore be able to identity novel associations.

p 5, line 103 - so the available data is continually coded in ICD8, even to the present?

line 112, 117 and elsehwere. What is the volume-adjusted Chi-square test? A little more detail would be helpful to readers.

line 123 what about associations requiring more than two birth defects - how does this method aggregate?

The section on methods contains a great deal of detail, more than is needed in the body of the manuscript. Consider placing some of this material in an appendix, and referring only to the terms and methods needed to describe the results in the methods section.

Table 1, p 8-9, the first two rows are repetitive across all columns. Place in the title or in a note at bottom of table instead.

p 9 - lots of VA-Chi2 values in text. What's required for one to be significant? 3.84??

Some of the material in the discussion belongs in results, for example lines 270-272 on p 11. The discussion is not the place to introduce findings not previously mentioned elsewhere.

This reviewer did not comment in detail on the discussion. But its hard to know whether these results 'seem to validate' the approach (p 14 line 336). To do this, it would be better to directly compare this method with other methods such as machine learning, discriminant or clustering analyses, etc, and show that it is similar or superior to those approaches.

6. PLOS authors have the option to publish the peer review history of their article (what does this mean?). If published, this will include your full peer review and any attached files.

Reviewer #1: Yes: Russell S. Kirby

---

## [Author Response · Author response to Decision Letter 0]

7 Apr 2020

We have incorporated your recommendations into the revised version of the manuscript. Thank you for your help.

---

## [Decision Letter · Decision Letter 1]

14 Apr 2020

PONE-D-19-35846R1

The Graph Theory: A new approach to analyze birth defect associations

PLOS ONE

Dear Dr López-Camelo,

Thank you for submitting your manuscript to PLOS ONE. After careful consideration, we feel that it has merit but does not fully meet PLOS ONE’s publication criteria as it currently stands. Therefore, we invite you to submit a revised version of the manuscript that addresses the points raised during the review process.

We would appreciate receiving your revised manuscript by May 29 2020 11:59PM. To enhance the reproducibility of your results, we recommend that if applicable you deposit your laboratory protocols in protocols.io, where a protocol can be assigned its own identifier (DOI) such that it can be cited independently in the future. For instructions see: http://journals.plos.org/plosone/s/submission-guidelines#loc-laboratory-protocols

We look forward to receiving your revised manuscript.

Kind regards,

Diego Raphael Amancio

Academic Editor

PLOS ONE

Reviewers' comments:

Reviewer's Responses to Questions

**Comments to the Author**

1. If the authors have adequately addressed your comments raised in a previous round of review and you feel that this manuscript is now acceptable for publication, you may indicate that here to bypass the “Comments to the Author” section, enter your conflict of interest statement in the “Confidential to Editor” section, and submit your "Accept" recommendation.

Reviewer #1: (No Response)

2. Is the manuscript technically sound, and do the data support the conclusions?

Reviewer #1: Partly

3. Has the statistical analysis been performed appropriately and rigorously? 

Reviewer #1: Yes

4. Have the authors made all data underlying the findings in their manuscript fully available?

Reviewer #1: Yes

5. Is the manuscript presented in an intelligible fashion and written in standard English?

Reviewer #1: Yes

6. Review Comments to the Author

Reviewer #1: The authors have revised the manuscript and addressed most major concerns. This reviewer still takes issue with the title - there is no 'the graph theory', but rather, a variety of graph-theoretic methods have been developed and used in a wide array of disciplines.

I strongly recommend that the title be changed to something like 'A Graph-Theoretic Approach to Analysis of Birth Defects Associations'.

Throughout the text, the authors should review how graph theory is described, for example in the next to last sentence of the abstract where it is also referred to as 'the graph theory'. Better to say '... validate graph-theoretic methods as ...'

It would also be important to consider how this method identifies potential unrecognized associations, considering study power (sample size), as well as inclusion and exclusion criteria.

7. PLOS authors have the option to publish the peer review history of their article (what does this mean?). If published, this will include your full peer review and any attached files.

Reviewer #1: Yes: Russell S. Kirby

---

## [Author Response · Author response to Decision Letter 1]

23 Apr 2020

Suggestions from the reviewer have been addressed in the revised version of the manuscript

---

## [Decision Letter · Decision Letter 2]

7 May 2020

A graph theory approach to analyze birth defect associations

PONE-D-19-35846R2

Dear Dr. López-Camelo,

We are pleased to inform you that your manuscript has been judged scientifically suitable for publication and will be formally accepted for publication once it complies with all outstanding technical requirements.

With kind regards,

Diego Raphael Amancio

Academic Editor

PLOS ONE

Additional Editor Comments (optional):

The authors should have this manuscript copyedited by someone familiar with scientific writing with English as a first language to improve the quality of the paper.

Reviewers' comments:

Reviewer's Responses to Questions

**Comments to the Author**

1. If the authors have adequately addressed your comments raised in a previous round of review and you feel that this manuscript is now acceptable for publication, you may indicate that here to bypass the “Comments to the Author” section, enter your conflict of interest statement in the “Confidential to Editor” section, and submit your "Accept" recommendation.

Reviewer #1: All comments have been addressed

2. Is the manuscript technically sound, and do the data support the conclusions?

Reviewer #1: Yes

3. Has the statistical analysis been performed appropriately and rigorously? 

Reviewer #1: Yes

4. Have the authors made all data underlying the findings in their manuscript fully available?

Reviewer #1: Yes

5. Is the manuscript presented in an intelligible fashion and written in standard English?

Reviewer #1: Yes

6. Review Comments to the Author

Reviewer #1: (No Response)

7. PLOS authors have the option to publish the peer review history of their article (what does this mean?). If published, this will include your full peer review and any attached files.

Reviewer #1: Yes: Russell S. Kirby

---

## [Editor Report · Acceptance letter]

12 May 2020

PONE-D-19-35846R2 

A graph theory approach to analyze birth defect associations 

Dear Dr. Lopez Camelo:

I am pleased to inform you that your manuscript has been deemed suitable for publication in PLOS ONE. Congratulations! Your manuscript is now with our production department. 

With kind regards,

on behalf of

Dr. Diego Raphael Amancio 

Academic Editor

PLOS ONE